# Effect of Processing Parameters on the Dynamic Characteristic of Material Extrusion Additive Manufacturing Plates

**Shijie Jiang** [1,*] **, Yinfang Shi** [1] **, Yannick Siyajeu** [1] **, Ming Zhan** [2] **, Chunyu Zhao** [1] **and Changyou Li** [1]

1   School of Mechanical Engineering and Automation, Northeastern University, Shenyang 110819, China; beyond_daisy@126.com (Y.S.); yannicksiyajeu@yahoo.fr (Y.S.); chyzhao@mail.neu.edu.cn (C.Z.); chyli@mail.neu.edu.cn (C.L.)
2   College of Information Science and Engineering, Northeastern University, Shenyang 110819, China; 19218793@163.com
*   Correspondence: jiangsj@me.neu.edu.cn



**Featured Application: With the development and deepening of this research, it will help improve the mechanical performance of ME components, and extend the applications of ME additive manufacturing technique.**

**Abstract:** Material extrusion (ME), an additive manufacturing technique, can fabricate parts almost without geometrical limitations. With the growing application of ME parts, especially in actual working conditions, the dynamic characteristics are needed to be studied to accurately determine their reliability. This study provides an experimental validation of the theoretical model for predicting the dynamic characteristics of ME plates fabricated with three different key processing parameters, i.e., extrusion width, layer height and build direction. The model is set up based on the bidirectional beam function combination method, and a series of experimental tests are performed. It is found that different processing parameters result in the material properties of the samples to vary, thus leading to different dynamic characteristics. Through the comparison between predictions and measurements, it is shown that the influencing trend of the processing parameters is predicted precisely. The theoretical model gives reliable predictions in dynamic characteristics of ME plates. The natural frequency discrepancy is below 13.4%, and the predicted mode shapes are the same as the measured ones. This present work provides theoretical basis and technical support for further research in improving the dynamic performance of ME products, and helps extend the applications of this technique.

**Keywords:** material extrusion; dynamic characteristic; theoretical model; experimental validation; processing parameters; material property

## 1. Introduction

Currently, increasing attention is being given to additive manufacturing techniques [1,2] compared to traditional processing techniques in order to promote competitiveness in the manufacturing market. Material extrusion (ME) technique, the term used here according to the standard of ISO/ASTM 52900:2015 to include the fused filament fabrication (FFF) or fused deposition modeling (FDM) technique, can fabricate products almost without geometrical limitations. It is an increasingly growing additive manufacturing technique in modern manufacturing industry, and the built end-use parts have been used in a wide range of applications, such as automotive, dentistry, electronic and aerospace

industry [3,4]. In this technique, the raw material filament is first melted in an extrusion head. It is then extruded out from the nozzle, which moves in a scheduled mode. The molten filament then cools, solidifies and bonds with the extruded material layer beneath it. A three-dimensional model or part entity is finally completed by repeating this procedure until the last layer.

During the past few years, extensive research has been carried out in studying various quality characteristics of ME products, such as mechanical property, surface roughness and build quality Hwang [5] developed new composite filaments mixing the acrylonitrile butadiene styrene (ABS) thermoplastic with copper and iron particles. The thermo-mechanical properties of the ME samples made of these filaments were investigated. It was found that the tensile strength was decreased with the loading of metal particles, but the thermal conductivity was improved. Torres [6] studied the effects of key processing parameters (the layer height, infill density and postprocessing heat-treatment time at 100 °C) on the material properties of ME polylactic acid (PLA) parts tested in torsion. It was concluded that to improve the fabricated parts' strength while preserving their ductility and reliability needs low levels of heat treatment. The layer height and infill density were important for optimizing strength, and ductility was significantly affected by infill and heat treatment. Tsouknidas [7] evaluated the shock mitigating properties of the polymeric structures fabricated by an ME device. The effect of layer height, infill pattern and density on the energy dissipation properties was examined. The results showed that the impact absorption capacity was influenced a lot by the infill density, whereas the effect of the other two factors was small. Li [8] studied the effects of layer height, deposition velocity and infill rate on ME parts' mechanical properties, and summarized the quantitative relationship between these processing parameters and the parts' tensile strength. The proposed theoretical model gave accurate predictions of the tensile strength, and it could reveal the work mechanism of these parameters. Rodríguez-Panes [9] compared the effect of layer height, infill density and build direction on the mechanical performance of PLA and ABS test samples, which included tensile yield stress, tensile strength, nominal strain at break, and modulus of elasticity. This work provided assistance in selecting proper filament materials for ME. Cuan-Urquizo et al. [10] thoroughly reviewed different experimental, analytical and computational studies for characterizing the mechanical properties of ME parts under different loading conditions. The applicability and limitations of these studies were summarized. The three approaches were recommended to be used together for accurately estimating the mechanical properties. The predictive tools, able to predict the mechanical performance, deformation and failures of ME parts, deserve further research. Durgun [11] investigated the selection of proper build directions and raster angles to achieve the optimum surface roughness and mechanical property of ME parts. The results showed that the direction has a more significant effect than the raster angle. The relationship between the surface roughness and mechanical properties was close. Kim [12] experimentally studied the effects of the gap between nozzle and substrate, filament feeding rate and printing speed on the ME parts' surface roughness. The width, height and cross-sectional shapes of the extrudate were examined. The results showed that these factors had significant effect on the built parts' surface roughness. Boschetto [13] developed a methodology able to improve the surface roughness of ME parts by computer numerical control machining. The proposed method was validated by a case study, achieving a great reduction of average roughness and a reliable uniformity of finished surfaces. Lalehpour [14] set up a reliable algorithm that could predict the surface roughness of ME parts with an accurate definition of the centerline taken into account. The method was then validated experimentally. With an ensemble learning algorithm considered, Li [15] proposed a data-driven predictive model for predicting the surface roughness of ME parts. Based on the condition monitoring data measured by multiple sensors of different types from an ME device, the predictive model is validated, and it was able to accurately predict the surface roughness of ME parts. Kestilä [16] combined atomic layer deposition (ALD) and ME technique to coat the aluminum oxide on the built small satellite propulsion parts. The benefits of the coating were studied in the context of in-space propulsion fluidics, with propellant flow properties taken into account. It was found that the parts' surface roughness was improved, leading to improved flow properties. Sun [17] investigated the mechanisms controlling the bonding formation between

the extruded filaments in ME process. The bonding quality was assessed in terms of measuring and analyzing the mesostructure and the degree of healing between the adjacent extrudate. The results showed that the build direction, liquefier temperature and convection coefficient had strong effects on the cooling temperature profile, as well as on the mesostructure and bonding strength of the built part. Gordeev [18] provided an efficient method to evaluate the quality of the parts built by ME. It was found that the wall permeability of the part was dependent on its geometric shape. The part's quality was primarily influenced by the filament feeding rate, wall geometry and G-code-defined wall structure. Optimizing these parameters could improve the quality (decrease in the porosity) and sealing properties of ME parts. Based on short beam shear tests, Caminero [19] evaluated the effect of layer height and fiber content on the interlaminar bonding performance of the fiber reinforced nylon composites manufactured by ME technique. It was observed that the layer height of nylon influences the interlaminar shear performance slightly. Fiber reinforced samples showed higher interlaminar shear performance than unreinforced ones, but the level of increase was moderate with the increase of fiber content. Narahara [20] used the atmospheric pressure plasma to carry out the hydrophilic treatment to the ME printing layer. This method could significantly improve the rupture strength and surface free energy (up by 75%) of ME parts, and thereby enhance their inter-layer bonding strength. Besides, more and more attentions have been paid to ME parts' accuracy levels and tolerances [21] to increase their industrial applications. Melenka [22] studied the dimensional accuracy of the parts built by an ME device under the same manufacturing condition. The analysis showed that the test specimens varied significantly from the nominal dimensions. Careful selection of processing parameters was suggested to optimize the dimensional accuracy. Boschetto [23] developed a design for manufacturing methodology able to improve the dimensional accuracy of ME parts. This approach, predicted the obtainable dimensional deviations in the model-design stage, provided compensations of the deviations generated in the fabrication stage. From the experiments, it was shown that the dimensional accuracy of the ME parts was improved significantly using this method. Turner [24] had done a literature review about the dimensional accuracy of ME parts. It was summarized that dimensional accuracy of the built parts was dependent upon the processing parameters and the properties of the feedstock filament properties. Particularly, the contours and layer height played key roles in determining the parts' dimensional accuracy. Nuñez [25] analyzed the dimensional accuracy of ME parts made of ABS-plus material in order to establish the dimensional tolerance ranges. The effect of two processing parameters (the infill density and layer height) on the dimensional accuracy was experimentally studied. It was concluded that the optimum configurations for the fabrication of ME parts with ABS-plus was set up. The infill density was the most influencing factor.

With the increasingly wider application of ME products in actual working conditions, the study in terms of vibration characteristics requires urgent attention. However, very few investigations have been focused on this research area, and the built parts' reliability can hardly be determined, especially in actual working conditions, such as machining, transportation, etc. Arivazhagan [26] used the DMA2980 equipment to conduct the frequency sweep test on ME samples from 1 to 100 Hz to determine their modulus, damping and viscosity values. The results illustrated that the strength of the part processed with the solid built style was higher than those fabricated with other built styles. Increasing temperature would increase their loss modulus, but decrease the viscosity values and storage modulus. Mohamed [27–29] used the same test method to investigate different processing parameters' effect on the dynamic elastic modulus of samples, and pointed out that modulus values would decrease when the build direction, raster angle or air gap increased. In addition, the numerical values of maximum glass transition temperature and dynamic modulus were determined by optimizing the processing parameters. Domingo-Espin [30] studied the effect of processing and testing parameters on the dynamic response of ME samples. Dynamic mechanical analysis (DMA) was used to determine the storage and loss modulus under an oscillating load. The results showed that the number of contours was the most influential parameter. Processing parameters (nozzle diameter, air gap and number of contours) could control the storage modulus of the manufactured parts, and testing parameters

(temperature, loading frequency, and amplitude) showed a great influence on their damping capacity. Mansour [31] performed a series of tests to investigate the dynamic behavior of ME parts made of polyethylene terephthalate glycol (PETG) and PETG reinforced with 20% carbon fibers. The results revealed that the loss factor and damping obtained from the cyclic compression and models tests dropped from 17.3% to 15.4% and 13.8% to 12.3%, respectively. Although the above studies were carried out under vibratory and cyclic conditions, the dynamics inherent characteristics of the samples were not taken into account, i.e., natural frequency and mode shape. Furthermore, the existing research mainly relies on experimental testing, and lacks theoretical basis and principle.

In this paper, a theoretical model was established and experimental research was performed to quantify the influence of different processing parameters on the dynamic inherent characteristics of ME plates. The sample preparations are introduced first, followed by the theoretical analysis and experimental tests. The samples' material properties are then provided, and the effect of three processing parameters on the dynamic characteristics is quantified through the comparison between predictions and measurements.

## 2. Materials and Methods

To investigate the dynamic characteristics of ME plates theoretically and experimentally, the fabrication of the samples was first introduced in this section, followed by the theoretical modeling and experimental analysis.

### 2.1. Sample Preparation

To determine the single effect of the processing parameter on tensile property, three kinds of dense-structure samples were fabricated by an ME device (D-force V2) according to ISO 527-2-2012 (International standard, plastics determination of tensile properties Part 2: test conditions for molding and extrusion plastics), as the schematic shown in Figure 1a. The first kind could be divided into three types, and they had different extrusion widths (a certain width of the filament extruded from the nozzle), which were separately 0.4, 0.3 and 0.2 mm. These samples were represented by $T_{E0.4\_i}$, $T_{E0.3\_i}$ and $T_{E0.2\_i}$. The second kind had another three types of samples built with different layer heights, which were 0.2, 0.15 and 0.1 mm. They were represented by $T_{L0.2\_i}$, $T_{L0.15\_i}$ and $T_{L0.1\_i}$. The third kind of samples also had three different types, whose build directions were X, Z and Oblique, individually represented by $T_{X0\_i}$, $T_{Z0\_i}$ and $T_{O0\_i}$. It is noted that only one processing parameter was set different from the original (or default) type of samples (i.e., $T_{E0.4\_i}$, $T_{L0.15\_i}$ and $T_{X0\_i}$) for each other type based on fractional factorial design. Details of the processing parameter settings are provided in Table 1. To ensure the reliability and repeatability of the measurements, more than ten samples of each type were fabricated (thus, $i > 10$).

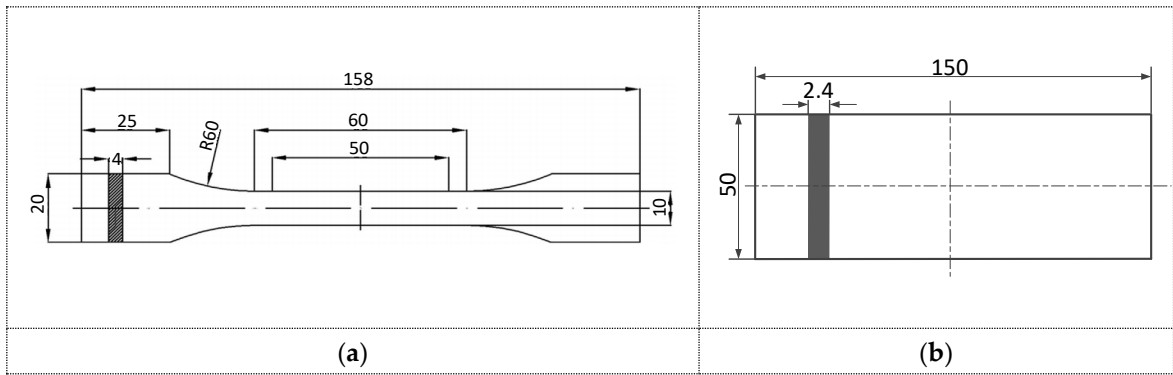

**Figure 1.** Two-dimensional drawing of the samples. (**a**) Tensile test sample and (**b**) dynamic test sample.

**Table 1.** The key processing parameter settings of the samples.

| Sample ($i > 10$) ($j = 1$–$5$) | Extrusion Width (mm) | Layer Height (mm) | Build Direction | Longitudinal Section | Number of Contours |
|---|---|---|---|---|---|
| $T_{E0.4\_i}$ $E_{0.4\_j}$ | 0.4 | 0.15 | X direction (0°) X direction (0°)  | Extrusion width | 3 |
| $T_{E0.3\_i}$ $E_{0.3\_j}$ | 0.3 | 0.15 | | Extrusion width | 4 |
| $T_{E0.2\_i}$ $E_{0.2\_j}$ | 0.2 | 0.15 | | Extrusion width | 6 |
| $T_{L0.2\_i}$ $L_{0.2\_j}$ | 0.4 | 0.2 | | Extrusion width | 3 |
| $T_{L0.15\_i}$ $L_{0.15\_j}$ | 0.4 | 0.15 | X direction (0°) | Extrusion width | 3 |
| $T_{L0.1\_i}$ $L_{0.1\_j}$ | 0.4 | 0.1 | | Extrusion width | 3 |
| $T_{Z0\_i}$ $Z_{0\_j}$ | 0.4 | 0.15 | Z direction (90°)  | Extrusion width | 3 |
| $T_{X0\_i}$ $X_{0\_j}$ | 0.4 | 0.15 | X direction (0°) | Extrusion width | 3 |
| $T_{O0\_i}$ $O_{0\_j}$ | 0.4 | 0.15 | Oblique direction (45°)  | Extrusion width | 3 |

For the dynamic characteristic, another three kinds of dense-structure samples were fabricated with the same ME device, as the schematic shown in Figure 1b. They were also fabricated based on fractional factorial design, with the three processing parameters (i.e., extrusion width, layer height and build direction) individually set to the values shown in Table 1. These samples were separately represented by $E_{0.4\_j}$, $E_{0.3\_j}$, $E_{0.2\_j}$, $L_{0.2\_j}$, $L_{0.15\_j}$, $L_{0.1\_j}$, $X_{0\_j}$, $Z_{0\_j}$ and $O_{0\_j}$, ($j = 1$–$5$). It is noted that $T_{E0.4\_i}$, $E_{0.4\_j}$, $T_{L0.15\_i}$, $L_{0.15\_j}$, $T_{X0\_i}$ and $X_{0\_j}$ were fabricated with the same processing parameters (the default values). All the samples mentioned above were made of polylactic acid (PLA [32], environmentally friendly material, having good workability and forming stability). Apart from only one single processing parameter set different for each type of samples, all the others were set the same, such as printing speed (moving speed of the printing head during the manufacturing process, set as 60 mm/s), extruder temperature (the temperature of the extruder to heat the filament to molten state, set as 200 °C), etc.

## 2.2. Theoretical Model

The 3D printing plate, composed of multiple layers of PLA fiber material, has the characteristic of orthogonal anisotropy. Figure 2 shows the geometric model of the rectangular plate, of which the length *a* was 130 mm, the width *b* was 50 mm and the total thickness *h* was 2.4 mm (the thickness of each single layer was the extrusion width of the extruded filament). Taking the central plane of the plate as the xoy plane, the three-dimensional coordinate system (O-xyz) was set up. The length, width and thickness directions are represented by x, y and z.

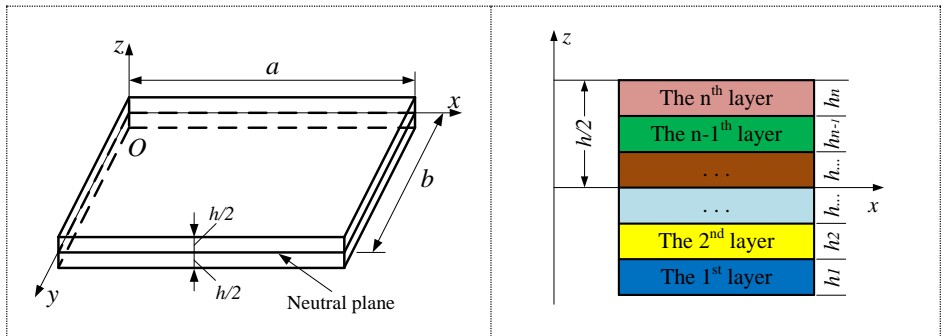

**Figure 2.** Mechanism graph of the material extrusion (ME) plate.

To model the plate, the following assumptions were made: (1) no delamination defect occurs inside the sample, the single layers are firmly bonded with each other, and no slip or relative displacement occurs between them. The sample can be regarded as an integral-structure plate, and there is no need to consider the interlayer coupling effect; (2) the bonding layer is thin, and no deformation occurs in itself, i.e., the deformation between the single-layer plates is continuous and (3) the plate is composed of multiple single-layer plates, but the total thickness is still consistent with the thin plate assumption. The modeling details are provided in the Appendix A.

## 2.3. Tensile Test

According to ISO 527-2-2012 standard, the tensile tests were carried out on the tensile test samples ($T_{E0.4\_i}$, $T_{E0.3\_i}$, $T_{E0.2\_i}$, $T_{L0.2\_i}$, $T_{L0.15\_i}$, $T_{L0.1\_i}$, $T_{Z0\_i}$, $T_{X0\_i}$ and $T_{O0\_i}$, $i > 10$) using the testing machine (Shimadzu EHF-EV200k2-040, Figure 3). The loading rate during the test was set to 5 mm/min. Since the material was PLA (the mechanical strength is much lower than steel [32]), the clamping force at both ends was set to only 5 MPa to avoid breaking the samples, which ensured the accuracy in the measurements. It is noted that the results of the samples broken in the middle part during the tensile test were considered useful, and five typical testing results were chosen and summarized to represent the stress–strain relationship of the dynamic test samples, which were fabricated with the sample processing parameters as the tensile test ones.

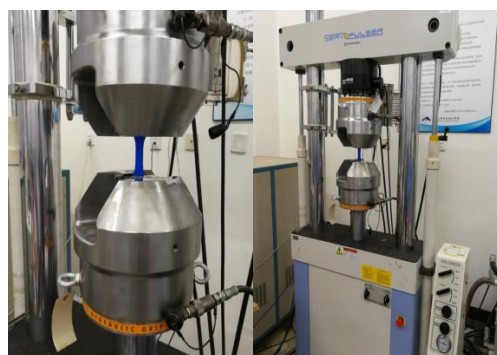

**Figure 3.** Tensile test machine (servo-hydraulic) and the sample.

### 2.4. Experimental Modal Analysis

To measure the dynamic characteristics of ME plates, modal tests were performed to the samples ($E_{0.4\_j}$, $E_{0.3\_j}$, $E_{0.2\_j}$, $L_{0.2\_j}$, $L_{0.15\_j}$, $L_{0.1\_j}$, $X_{0\_j}$, $Z_{0\_j}$ and $O_{0\_j}$, $j = 1$–5). Figure 4 shows the testing system that mainly includes an impact hammer (PCB 086C01, with the sensitivity being 11.2 mV/N), a data acquisition card (NI USB 4431, with four analog input channels and one analog output channel) and a lightweight accelerometer (PCB 352C33, with the sensitivity being 103.5 mV/g). The plates were fixed with the clamping device and under the cantilever boundary conditions, with the length of the clamping area being 20 mm. The excitation point was about 10 mm away from the bottom, as shown in Figure 4. To ensure the accuracy of the measurements, the accelerometer was successively fixed to the measuring points on the plate with a large vibration response (the top and the middle positions of the sample, avoiding the node).

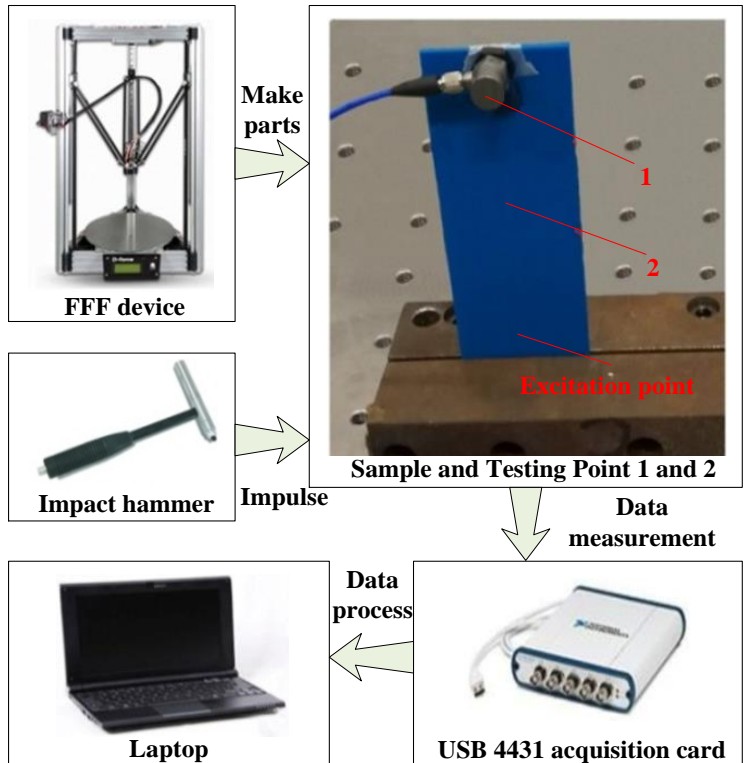

**Figure 4.** Testing system for the determination of the dynamic characteristics of ME plates.

In the experiments, the impulse excitation was first applied to the samples by the hammer. The vibration responses of the samples due to the excitation were then measured by the accelerometer, with the sampling rate set as 2000 Hz. At the same time, the real-time excitation signals and response signals were collected through the data acquisition card. In order to obtain the mode shape of the samples, MISO (multiple-input single-output) method was used to identify the parameter, i.e., the accelerometer was fixed at a position with a large vibration response, and the other measuring positions were excited separately. It is noted that no filtering was implemented as no specific frequency band needed to be filtered out. For each sample, 10 groups of tests were completed which further ensures the accuracy of the measurements (350 groups of measured data obtained in total).

### 3. Results and Discussion

In this section, the material properties of the ME samples were provided, followed by the corresponding scanning electron microscope (SEM) analysis. The effect of three processing parameters

(extrusion width, layer height and build direction) on the sample's dynamic characteristics was then detailed.

### 3.1. Material Property of the Sample

The samples were weighted using the electrical analytical balance (FA2004), of which the precision was 0.1 mg. Together with the volume of the dynamic test plates, the density of each sample was determined. Based on Hooke's law, the elastic modulus of the samples was obtained. According to the equation $G = E/2(1 + v)$, the ME plates' shear elastic modulus could be determined, where $v$ is Poisson ratio, and $v = 0.36$ [32]. Table 2 summarizes the average measured material properties of each type of dynamic test samples. Since $E_{0.4}$, $L_{0.15}$ and $X_0$ were fabricated with the same processing parameters, they had the same properties.

**Table 2.** Average material properties of material extrusion (ME) samples.

| Order | Sample | Average Elastic Modulus (MPa) | Average Shear Modulus (MPa) | Average Density (g/cm$^3$) | Poisson's Ratio |
|---|---|---|---|---|---|
| 1 | $E_{0.4}$, $L_{0.15}$ and $X_0$ | 18.8 | 6.91 | 0.92 | |
| 2 | $E_{0.3}$ | 18.5 | 6.80 | 0.91 | |
| 3 | $E_{0.2}$ | 18.0 | 6.62 | 0.91 | 0.36 |
| 4 | $L_{0.2}$ | 19.3 | 7.10 | 0.92 | |
| 5 | $L_{0.1}$ | 18.1 | 6.65 | 0.91 | |
| 6 | $Z_0$ | 17.1 | 6.29 | 0.90 | |
| 7 | $O_0$ | 17.6 | 6.47 | 0.93 | |

It can be seen that the wider the extrusion width is, the larger the material property of the corresponding samples will be. This can be explained that the bonding area between adjacent extruded filaments will be decreased when the extrusion width decreases, as seen from the comparison between Figure 5a,b, leading to the deterioration of bonding strength. There would be more thermal cycles and more non-uniform thermal gradients, leading to more distortions, deformations and non-uniformities of the extruded filament (as shown from Figure 6a,b), which is a major reason for the bonded layers' poor quality [27], and thus worse bonding quality. In addition, decreasing extrusion width would increase the bonding gap between the filaments of the cross section, generating more defects such as delamination, internal pores and void structures. Therefore, decreasing extrusion width will deteriorate the material property of the ME product.

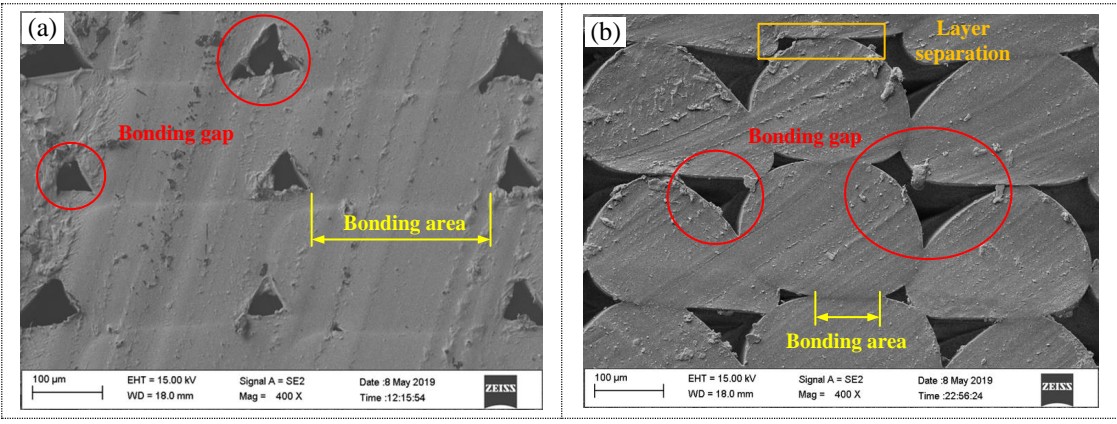

**Figure 5.** *Cont.*

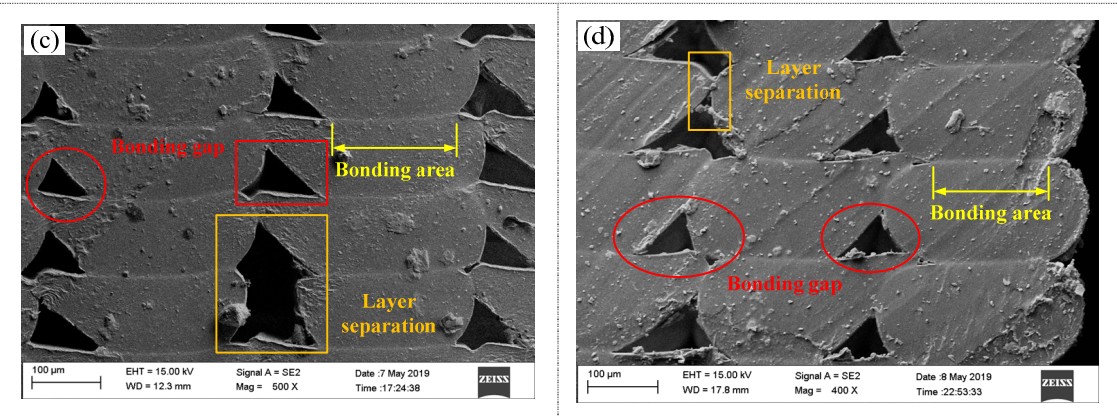

**Figure 5.** Scanning electron microscope (SEM) micrograph of the sample's cross section. (**a**) Sample $E_{0.4}$ (or $L_{0.15}$, $X_0$); (**b**) Sample $E_{0.2}$; (**c**) Sample $L_{0.1}$ and (**d**) Sample $Z_0$.

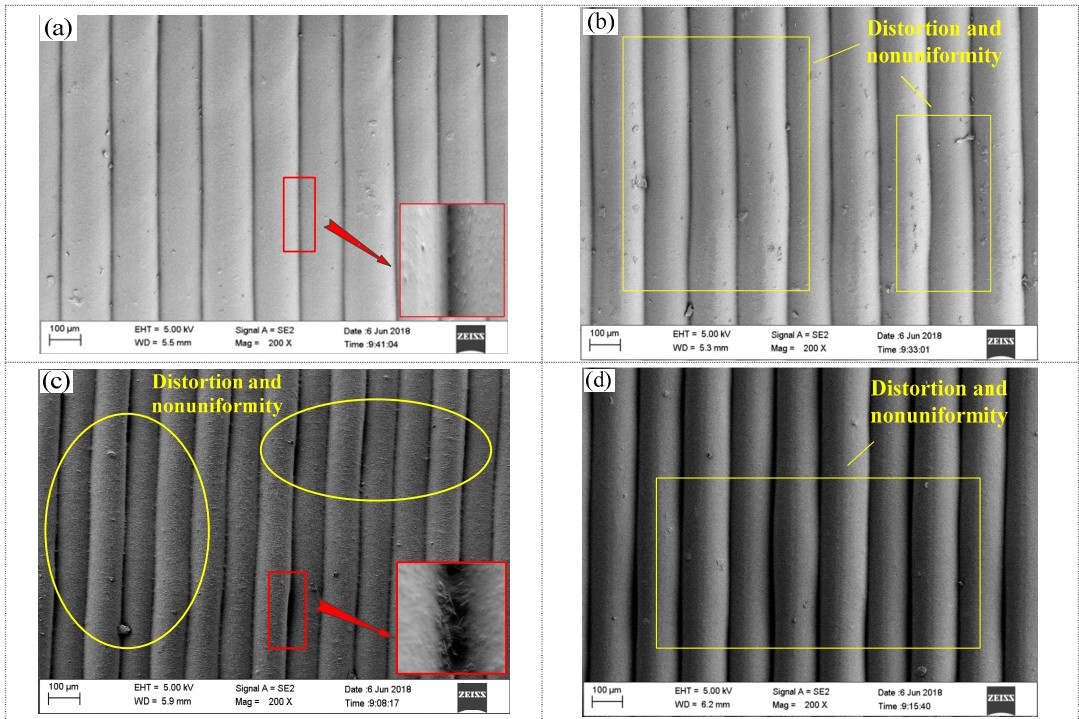

**Figure 6.** Surface SEM observation of the sample. (**a**) Sample $E_{0.4}$ (or $L_{0.15}$, $X_0$); (**b**) Sample $E_{0.2}$; (**c**) Sample $L_{0.1}$ and (**d**) Sample $Z_0$.

Decreasing layer height can reduce the samples' material property. When layer height is decreased, the number of layers needed to fabricate the sample will increase, resulting in the increase of bonding layers and thermal cycles, which further lead to more distortions and non-uniformities in the sample, as shown in Figure 6c. The bonding quality and strength between extruded filaments became worse. In addition, the defects like porosity, gap, separation, etc. would increase in the sample, as shown in Figure 5c. Therefore, the material property of the sample will be decreased when the layer height is reduced.

The samples built in the X direction had larger material properties than those built in other directions. As the build direction changed from X to Z, more layers were required to manufacture the part, resulting in more bonding layers, thermal cycles and distortions (Figure 6d, the distortion occurred due to high non-uniform temperature gradient between the bottom and the top layers during the printing). The bonding quality and strength would be reduced (see Figure 5d), and thus the material properties of the built part were deteriorated.

### 3.2. Effect of Extrusion Width on the Dynamic Property

Figure 7 presents the average measured frequency response curves of a set of ME plates fabricated with different extrusion widths separately being 0.4, 0.3 and 0.2 mm. It is noted that no filtering was implemented as no specific frequency band needed to be filtered out. By analyzing and identifying the peaks of the curves, the natural frequencies of each sample could be preliminarily determined. The results are detailed in Table 3. It can be seen that Sample $E_{0.4}$ had the lowest values of resonant response (occurring due to the impulse excitation), meaning that the anti-vibration capability was the best compared with Sample $E_{0.3}$ and $E_{0.2}$. Therefore, the sample's anti-vibration capability would be improved if the extrusion width was increased.

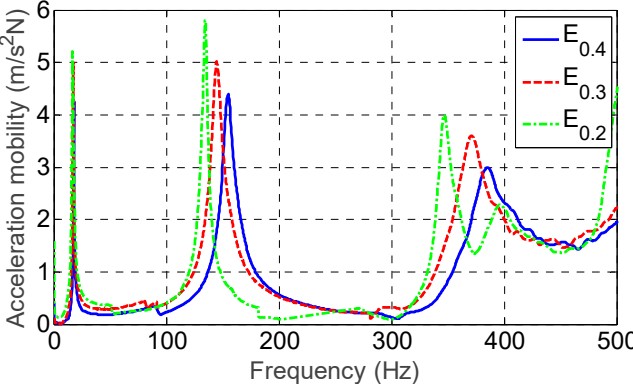

**Figure 7.** The average measured frequency response of the samples fabricated with different extrusion width.

**Table 3.** The resonant response values of the samples fabricated with different extrusion widths.

| Sample | Average Resonant Response (m/s² N) | | |
|---|---|---|---|
| | 1st Order | 2nd Order | 3rd Order |
| $E_{0.4}$ | 4.3 | 4.4 | 3.0 |
| $E_{0.3}$ | 5.0 | 5.0 | 3.6 |
| $E_{0.2}$ | 5.2 | 5.8 | 4.0 |

Table 4 summarizes the effect of different extrusion widths on the values of predicted and measured dynamic inherent characteristics. It can be seen from the comparison that the predictions agreed well with the measurements. The influencing trend of different extrusion width was predicted precisely. The discrepancy between predictions and measurements in natural frequency was in the range of 0.7–11%. The predicted mode shapes were the same as those of the measurements. The 1st order mode shape was the 1st order bending vibration, the 2nd order one was the 1st order torsional vibration and the 3rd order one was the 2nd order bending vibration. Therefore, the theoretical modeling based on the bidirectional beam function combination method could give reliable predictions on the inherent characteristics of ME plates. In addition, larger extrusion width mainly led to the increase in the material properties, i.e., elastic modulus, shear modulus and density, which would enhance the strain (or structural stiffness), potential and kinetic energy in the plates according to Equations (A12), (A14) and (A15). It eventually increased the natural frequency. For example, the values of material properties of Sample $E_{0.4}$ were larger than those of $E_{0.3}$ and $E_{0.2}$ as shown in Table 2, the corresponding natural frequency was the highest, with the first three measured natural frequencies being 18 Hz, 154.9 Hz and 384.8 Hz. The corresponding predicted natural frequencies were 19.3 Hz, 170.1 Hz and 396.4 Hz. Since the structural characteristics were not changed with various extrusion widths, the mode shapes were kept the same.

**Table 4.** The effect of different extrusion width on the samples' inherent characteristics.

| Item | Samples | Order | | |
| --- | --- | --- | --- | --- |
| | | 1st | 2nd | 3rd |
| Average measured natural frequency *A* (Hz) | $E_{0.4}$ | 18.0 | 154.9 | 384.8 |
| | $E_{0.3}$ | 17.1 | 145.6 | 371.1 |
| | $E_{0.2}$ | 16.1 | 134.2 | 346.0 |
| Average predicted natural frequency *B* (Hz) | $E_{0.4}$ | 19.3 | 170.1 | 396.4 |
| | $E_{0.3}$ | 18.5 | 161.6 | 373.6 |
| | $E_{0.2}$ | 17.0 | 147.6 | 334.4 |
| Error (%) $|B - A|/A$ | $E_{0.4}$ | 7.2 | 9.8 | 3.0 |
| | $E_{0.3}$ | 8.2 | 11.0 | 0.7 |
| | $E_{0.2}$ | 5.6 | 10.0 | 3.4 |
| Measured mode shape | | | | |
| Predicted mode shape | | | | |

## 3.3. Effect of Layer Height on the Dynamic Property

Figure 8 shows the effect of different layer height (0.1, 0.15 and 0.2 mm) on the frequency response of ME plates, with no filtering implemented. The results are detailed in Table 5. It could be seen that Sample $L_{0.1}$ had the highest values of resonant response (occurring due to the impulse excitation), meaning that the anti-vibration capability was worse than Sample $L_{0.15}$ and $L_{0.2}$. Therefore, the sample's anti-vibration capability would be improved if the layer height was increased.

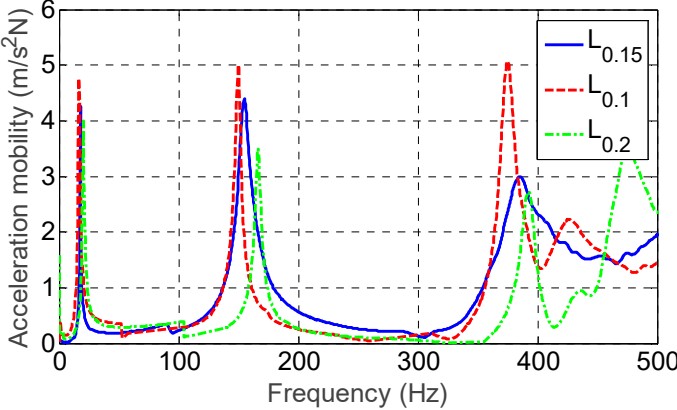

**Figure 8.** The average measured frequency response of the samples fabricated with different layer height.

**Table 5.** The resonant response values of the samples fabricated with different layer heights.

| Sample | Average Resonant Response (m/s$^2$ N) | | |
| --- | --- | --- | --- |
| | 1st Order | 2nd Order | 3rd Order |
| $L_{0.1}$ | 4.8 | 5.0 | 5.1 |
| $L_{0.15}$ | 4.3 | 4.4 | 3.0 |
| $L_{0.2}$ | 4.0 | 3.5 | 2.7 |

The corresponding dynamic inherent characteristics are detailed in Table 6. It could be seen that the predicted natural frequencies and mode shapes agreed well with the measured data. The influencing trend of different layer heights was predicted precisely. The discrepancy of natural frequency was between 0.9% and 13.3%, the predicted mode shapes were the same as the measured ones. These confirmed the reliability of the theoretical modeling. Besides, different layer height caused the changes of elastic modulus, shear modulus and density of the samples, and eventually influenced the natural frequency. Since Sample $L_{0.2}$ had the largest values of material property (shown in Table 2), the corresponding natural frequency was the highest. The first three measured natural frequencies were 20.1 Hz, 167.1 Hz and 392.3 Hz, with the predicted values being 21.3 Hz, 189.3 Hz and 434.3 Hz. Since different extrusion widths did not change the structural characteristics of the samples, the mode shapes were kept the same.

**Table 6.** The effect of different layer height on samples' inherent characteristics.

| Item | Samples | Order | | |
|---|---|---|---|---|
| | | 1st | 2nd | 3rd |
| Average measured natural frequency *A* (Hz) | $L_{0.1}$ | 16.6 | 148.9 | 375.1 |
| | $L_{0.15}$ | 18.0 | 154.9 | 384.8 |
| | $L_{0.2}$ | 20.1 | 167.1 | 392.3 |
| Average predicted natural frequency *B* (Hz) | $L_{0.1}$ | 17.0 | 150.2 | 342.7 |
| | $L_{0.15}$ | 19.3 | 170.1 | 396.4 |
| | $L_{0.2}$ | 21.3 | 189.3 | 434.3 |
| Error (%) $\lvert B - A \rvert / A$ | $L_{0.1}$ | 2.4 | 0.9 | 8.6 |
| | $L_{0.15}$ | 7.2 | 9.8 | 3.0 |
| | $L_{0.2}$ | 6.0 | 13.3 | 10.7 |
| Measured mode shape | |  |  |  |
| Predicted mode shape | |  |  |  |

### 3.4. Effect of Build Direction on the Dynamic Property

Figure 9 provides the frequency response of the ME plates fabricated with different build directions, which were X, Z and Oblique directions. No filtering was implemented as no specific frequency band needed to be filtered out. Table 7 details the data. Since Sample $X_0$ had the lowest values of resonant response, its anti-vibration capability was the best compared with Sample $O_0$ and $Z_0$. Therefore, the sample's anti-vibration capability would be improved if it was built in the X direction.

Table 8 details the effect of different build directions on the inherent characteristics of ME plates. It could be seen that the influencing trend of different build directions was predicted precisely. The difference between predictions and measurements in natural frequency was from 2.3% to 13.4% and the predicted mode shapes were the same as the measurements. These further confirmed that the theoretical modeling was reliable in predicting the plate's dynamic inherent characteristics. Furthermore, different build directions would change the samples' elastic modulus, shear modulus and density, and further influenced the corresponding natural frequencies. For example, the material property values of Sample $Z_0$ were the smallest among $Z_0$, $X_0$ and $O_0$, as listed in Table 2, the corresponding natural frequency was the lowest, with the first three measured natural frequencies separately being 15.2 Hz, 130.3 Hz and 323 Hz. The predicted data was 14.1 Hz, 112.8 Hz and 290.9 Hz.

Since different build directions did not change the structural characteristics of the samples, the mode shapes were kept the same.

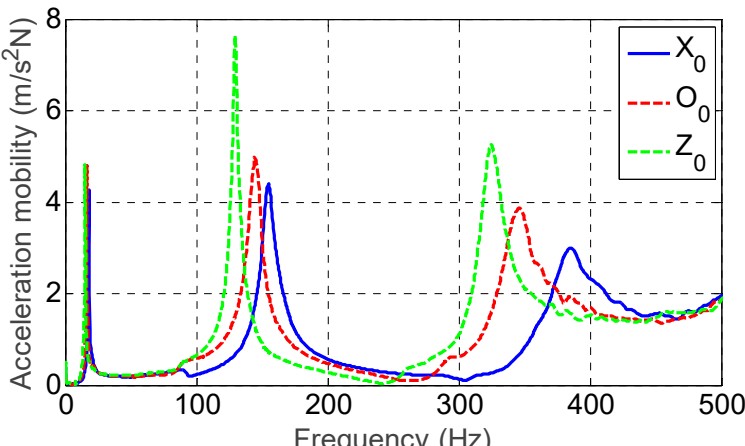

**Figure 9.** The average measured frequency response of the samples fabricated with different build directions.

**Table 7.** The resonant response values of the samples fabricated with different build directions.

| Sample | Average Resonant Response ($m/s^2$ N) | | |
|---|---|---|---|
| | **1st Order** | **2nd Order** | **3rd Order** |
| $X_0$ | 4.3 | 4.4 | 3.0 |
| $O_0$ | 4.8 | 4.9 | 3.9 |
| $Z_0$ | 4.8 | 7.7 | 5.3 |

**Table 8.** The effect of different build directions on the samples' inherent characteristics.

| Item | Samples | Order | | |
|---|---|---|---|---|
| | | **1st** | **2nd** | **3rd** |
| Average measured natural frequency $A$ (Hz) | $Z_0$ | 15.2 | 130.3 | 323 |
| | $X_0$ | 17.8 | 155.1 | 387.6 |
| | $O_0$ | 16.7 | 145.5 | 346.5 |
| Average predicted natural frequency $B$ (Hz) | $Z_0$ | 14.1 | 112.8 | 290.9 |
| | $X_0$ | 19.3 | 170.1 | 396.4 |
| | $O_0$ | 16.1 | 132.2 | 311.7 |
| Error (%) $|B - A|/A$ | $Z_0$ | 7.2 | 13.4 | 9.9 |
| | $X_0$ | 8.4 | 9.7 | 2.3 |
| | $O_0$ | 3.6 | 9.1 | 10.0 |
| Measured mode shape | | | | |
| Predicted mode shape | | | | |

## 4. Conclusions

Combined theory with experiment, this paper quantified the effect of extrusion width, layer height and build direction on the dynamic characteristics of material extrusion (ME) additive manufacturing

plates under cantilever boundary conditions. The ME plate's material properties will be deteriorated as the extrusion width decreases. Increasing layer height will improve the corresponding material property. When the build direction changes from X to Z, the material property of the sample will be reduced. Compared with the measurements, the theoretical model set up based on the bidirectional beam function combination method was validated, and it could give reliable predictions in the inherent characteristic of ME plates. The influencing trend of the processing parameters was predicted precisely. The natural frequency discrepancy was below 13.4%, and the predicted mode shapes were similar to the measured ones. In addition, different processing parameters, i.e., extrusion width, layer height and build direction, mainly led to the differences in samples' elastic modulus, shear modulus and density. When these values were greater, the anti-vibration capability became better and the corresponding natural frequency was higher. As the structural characteristics were not changed with varying processing parameters, there was no change in the mode shape. Besides, the material used here was polylactic acid (PLA), which is just a type of common material. The modeling and testing methods mentioned in this present work could also be applied to other materials. This paper provided a theoretical basis and technical support for further research in improving the dynamic performance of ME products.

**Author Contributions:** Conceptualization, S.J.; data curation, Y.S. and Y.S.; formal analysis, S.J., Y.S. and Y.S.; funding acquisition, S.J.; investigation, Y.S., Y.S. and C.L.; methodology, S.J. and C.Z.; project administration, S.J. and M.Z.; resources, S.J., C.L. and C.Z.; software, Y.S.; supervision, S.J. and M.Z.; validation, S.J., Y.S. and Y.S.; writing—original draft preparation, Y.S. and Y.S.; writing—review and editing, S.J. and C.Z.

**Funding:** The work described was supported by National Natural Science Foundation of China (51705068) and the fundamental research funds for the central universities (N180703009).

**Acknowledgments:** The first author is grateful for the technical support from Hui Li, who is working in the same academic institute as the author.

**Conflicts of Interest:** The authors declare no conflicts of interest.

## Appendix A

According to the small deflection theory of thin plates (Kirchoff hypothesis), the bidirectional beam function combination method [33] could be used to approximate the real mode shape of the plate under any boundary conditions (i.e., simply supported, fixed, free or any combinations), and further to analyze the inherent characteristics.

The plate's mode shape can be expressed as

$$W_p(x,y) = \sum\nolimits_{i=1}^{n} \sum\nolimits_{j=1}^{n} a_{ij} X_i(x) Y_j(y), \tag{A1}$$

where $X_i(x)$ is the $i^{\text{th}}$ order mode shape function meeting the boundary condition in $x$ direction, $Y_j(y)$ is the $j^{\text{th}}$ order mode shape function meeting the boundary condition in $y$ direction, $a_{ij}$ is an undetermined coefficient, which is utilized to adjust the combination of beam functions in different orders so as to make the mode shape function closer to the real mode shape of the rectangular plate.

The general expression of beam mode shapes is

$$Y(x) = A \sin kx + B \cos kx + C \text{sh} kx + D \text{ch} kx. \tag{A2}$$

The x-axis direction is a free boundary condition, considering $X_i(x)$ as the $n^{\text{th}}$ order's mode shape function of a free-free beam, the boundary condition needs to satisfy

When $x = 0$ and $x = a$,

$$EJ\frac{\text{d}^2 Y}{\text{d}x^2} = 0, \quad \frac{\text{d}}{\text{d}x}\left[EJ\frac{\text{d}^2 Y}{\text{d}x^2}\right] = 0. \tag{A3}$$

Substitute Equation (A2) into Equation (A3), the expression of each order mode shape of the free-free beam is

$$
\begin{aligned}
X_1 &= 1 \\
X_2 &= \sqrt{3(1 - 2X/a)}, \\
X_i &= \operatorname{ch}\tfrac{k_i X}{a} + \cos\tfrac{k_i X}{a} - \alpha_i\left(\operatorname{sh}\tfrac{k_i X}{a} + \sin\tfrac{k_i X}{a}\right),
\end{aligned}
\tag{A4}
$$

where $k_i$ is the coefficient related to beam frequency, $k_i^4 = \omega^2(\rho A/EJ)$, $\alpha_i$ is the beam function coefficient and $\alpha_i = (\operatorname{ch} k_i a - \cos k_i a)/(\operatorname{sh} k_i a - \sin k_i a)$, ($i = 3, 4, 5 \ldots$). The 1st order ($X_1$) shows the movement of rigid body; the 2nd order ($X_2$) shows the torsion of rigid body; starting from the 3rd order, the vibration mode is caused by the beam deformation.

In y-axis direction, the plate can be seen as a fixed-free beam, the boundary condition needs to satisfy

When y = 0,

$$
Y = 0, \quad \frac{dY}{dx} = 0.
\tag{A5}
$$

When y = a,

$$
EJ\frac{d^2 Y}{dx^2} = 0, \quad \frac{d}{dx}\left[EJ\frac{d^2 Y}{dx^2}\right] = 0.
\tag{A6}
$$

Substitute Equation (A2) into Equations (A5) and (A6), the expression of each order's mode shape function of a fixed-free beam is

$$
Y_i = \operatorname{ch}\frac{k_j y}{b} - \cos\frac{k_j y}{b} - \alpha_j\left(\operatorname{sh}\frac{k_j y}{b} - \sin\frac{k_j y}{b}\right),
\tag{A7}
$$

where $k_j$ is the same as $k_i$ (mentioned above) and $\alpha_j = \left(\operatorname{ch} k_j b - \cos k_j b\right)/\left(\operatorname{sh} k_j b - \sin k_j b\right)$, ($i = 3, 4, 5 \ldots$). The frequency coefficient [33] is detailed in Table A1.

Substitute Equations (A4) and (A7) into Equation (A1), the mode shape function of the ME plate under cantilever boundary condition is given by

$$
\begin{aligned}
W_p = \ & X_1 \cdot Y_1 \cdot a_{11} + X_1 \cdot Y_2 \cdot a_{12} + X_1 \cdot Y_3 \cdot a_{13} \\
& + X_2 \cdot Y_1 \cdot a_{21} + X_2 \cdot Y_2 \cdot a_{22} + X_2 \cdot Y_3 \cdot a_{23} \\
& + X_3 \cdot Y_1 \cdot a_{31} + X_3 \cdot Y_2 \cdot a_{32} + X_3 \cdot Y_3 \cdot a_{33} + \ldots
\end{aligned}
\tag{A8}
$$

where

$$
\left\{
\begin{aligned}
X_1 =\ & 1 \\
X_2 =\ & \sqrt{3(1 - 2x/a)} \\
X_3 =\ & \operatorname{ch}\tfrac{4.73x}{a} + \cos\tfrac{4.73x}{a} - \\
& \tfrac{\operatorname{ch} k_i a - \cos k_i a}{\operatorname{sh} k_i a - \sin k_i a}\left(\operatorname{sh}\tfrac{4.73x}{a} + \sin\tfrac{4.73x}{a}\right) \\
Y_1 =\ & \operatorname{ch}\tfrac{1.875y}{b} - \cos\tfrac{1.875y}{b} - \\
& \tfrac{\operatorname{ch} 1.875b - \cos 1.875b}{\operatorname{ch} 1.875b - \cos 1.875b}\left(\operatorname{sh}\tfrac{1.875y}{b} - \sin\tfrac{1.875y}{b}\right) \\
Y_2 =\ & \operatorname{ch}\tfrac{4.694y}{b} - \cos\tfrac{4.694y}{b} - \\
& \tfrac{\operatorname{ch} 4.694b - \cos 4.694b}{\operatorname{ch} 4.694b - \cos 4.694b}\left(\operatorname{sh}\tfrac{4.694y}{b} - \sin\tfrac{4.694y}{b}\right) \\
Y_3 =\ & \operatorname{ch}\tfrac{4.694y}{b} - \cos\tfrac{4.694y}{b} - \tfrac{\operatorname{ch} 7.855b - \cos 7.855b}{\operatorname{ch} 7.855b - \cos 7.855b}\left(\operatorname{sh}\tfrac{7.855y}{b} - \sin\tfrac{7.855y}{b}\right)
\end{aligned}
\right. .
$$

**Table A1.** The coefficient of beam frequency.

| Boundary Conditions | | $\alpha_i$ | | |
| --- | --- | --- | --- | --- |
| $x=0$ | | $i=1$ | $i=2$ | $i=3$ |
| Free | Free | 0 | 0 | 4.73 |
| Fixed | Free | 1.875 | 4.694 | 7.855 |

Since the accurate theoretical results can hardly be determined by the analytical method, the Ritz method was adopted in this paper. It is an approximate solution method according to the energy variation principle, which could obtain the results by transforming functional extremum problem to the extremum problem of multivariate function. Due to the vibration excitation, the plate would be deformed, producing stress and strain. Therefore, there is deformation energy varying with time existing in the plate. The strain energy density can be expressed by

$$W_p = \frac{1}{2}\left(\sigma_X \varepsilon_X + \sigma_Y \varepsilon_Y + \tau_{xy}\gamma_{xy}\right). \tag{A9}$$

Set the deformation function of central plane as $w_0(x, y, t)$, the displacement function of any point in the plate can be given by

$$
\begin{aligned}
u(x,\ y, z, t) &= -z\frac{\partial w_0(x,y,t)}{\partial x}, \\
v(x,\ y, z, t) &= -z\frac{\partial w_0(x,y,t)}{\partial y}.
\end{aligned}
\tag{A10}
$$

According to geometric equation, the strain of any point in the plate can be obtained as

$$
\begin{aligned}
\varepsilon_x &= \frac{\partial u}{\partial x} = -z\frac{\partial^2 w}{\partial x^2} = -z\kappa_x, \\
\varepsilon_y &= \frac{\partial v}{\partial y} = -z\frac{\partial^2 w}{\partial y^2} = -z\kappa_y, \\
\gamma_{xy} &= \frac{\partial u}{\partial y} + \frac{\partial v}{\partial x} = 2z\frac{\partial^2 w}{\partial x \partial y} = -2z\kappa_{xy},
\end{aligned}
\tag{A11}
$$

where $\kappa_x$, $\kappa_y$ and $\kappa_{xy}$ are the curvature and distortion of the central plane.

According to stress–strain relationship, the stress of any point in the plate could be expressed by

$$
\begin{aligned}
\sigma_x &= \frac{E}{1-v^2}\left(\varepsilon_x + v\varepsilon_y\right) = -\frac{E}{1-v^2}z\left(\frac{\partial^2 w}{\partial x^2} + v\frac{\partial^2 w}{\partial x^2}\right) \\
\sigma_y &= \frac{E}{1-v^2}\left(\varepsilon_y + v\varepsilon_y\right) = -\frac{E}{1-v^2}z\left(\frac{\partial^2 w}{\partial y^2} + v\frac{\partial^2 w}{\partial x^2}\right), \\
\tau_{xy} &= G\gamma_{xy} = -2G\,z\frac{\partial^2 w}{\partial x \partial y},
\end{aligned}
\tag{A12}
$$

where $v$ is poisson's ratio, E is elastic modulus and G is shear modulus.

Substituting Equations (A11) and (A12) into Equation (A10) gives

$$
\begin{aligned}
W_p &= \frac{1}{2}\left[\frac{1}{E}\left(\sigma_x^2 + \sigma_y^2 - 2v\sigma_x\sigma_y\right) + \frac{1}{G}\tau_{xy}^2\right] \\
&= \frac{Ez^2}{2(1-v^2)}\left[\left(\frac{\partial^2 w}{\partial x^2}\right)^2 + \left(\frac{\partial^2 w}{\partial y^2}\right)^2 + 2(1-v)\left(\frac{\partial^2 w}{\partial x \partial y}\right)^2\right].
\end{aligned}
\tag{A13}
$$

Therefore, the potential energy of the plate is given by

$$U = \frac{D}{2}\iint_A \left\{\left(\frac{\partial^2 w}{\partial x^2} + \frac{\partial^2 w}{\partial y^2}\right)^2 - 2(1-v)\left[\left(\frac{\partial^2 w}{\partial x^2}\right)\left(\frac{\partial^2 w}{\partial y^2}\right) - \left(\frac{\partial^2 w}{\partial x \partial y}\right)^2\right]\right\}dxdy. \tag{A14}$$

The kinetic energy of the plate is given by

$$T = \frac{\rho h}{2}\iint_A \left(\frac{\partial^2 W}{\partial t^2}\right)^2 dxdy. \tag{A15}$$

Substitute the expression of displacement $W(x, y, t) = W(x, y) \sin(wt + \varphi)$ into Equations (A14) and (A15), and then into the variational formula based on Hamilton principle ($\delta \int_{t_1}^{t_2} (T - U) dt = 0$), the time integral yields

$$\delta(U_{max} - T_{max}) = 0. \tag{A16}$$

$$U_{max} = \frac{D}{2} \iint_A \left\{ \left( \frac{\partial^2 w}{\partial x^2} + \frac{\partial^2 w}{\partial y^2} \right)^2 - 2(1 - v) \left[ \left( \frac{\partial^2 w}{\partial x^2} \right) \left( \frac{\partial^2 w}{\partial y^2} \right) - \left( \frac{\partial^2 w}{\partial x \partial y} \right)^2 \right] \right\} dx dy. \tag{A17}$$

$$T_{max} = \frac{W^2 \rho h}{2} \iint_A W^2 dx dy. \tag{A18}$$

Substitute the mode shape function (Equation (A8)) into Equations (A17) and (A18), and let the partial derivative of $U_{max} - T_{max}$ in terms of the undetermined coefficient $\alpha_{ij}$ equal 0, that is

$$\frac{dU_{max}}{d\alpha_{ij}} - \frac{dT_{max}}{d\alpha_{ij}} = 0. \tag{A19}$$

The linear algebraic equation for $\alpha_{ij}$ is thus obtained. Set the coefficient determinant of the equation to be 0, the circular frequency $\omega$ can be determined. According to $f = \omega/2\pi$, the natural frequency of the ME plate can be obtained.

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
