# Peer review of "Effect of Processing Parameters on the Dynamic Characteristic of Material Extrusion Additive Manufacturing Plates"

_applsci, doi:10.3390/app9245345_

Round 1
Reviewer 1 Report
This paper offers an interesting study of the influence of process parameters on mechanical properties of parts manufactured by FDM 3D printing.
Although the subject of study has been faced by other authors previously, the rigor of the study, the methodology and tests carried out by the authors can provide additional information to progress in the state of the art.
Some aspects could be improved previously to acceptance:
- Keywords do not reflect well the scope of the research.
- Some references are not necessary. For example, references [3-5] for explaining the FDM or FFF process. Nowadays, it is a well-known process.
- Other references are mentioned as a unique set ([6-10], [11-14], [15-18], [19-21]), without commenting specific issues for each one. It is accepted that each reference included in a paper should be commented individually, indicating specific issues of each one.
- In page 2, references to additive manufacturing process with Ti6Al4V are not adequate in this case. Metal printing is very different to FFF process. It is not comparable and must be suppressed from the text.
- In the theoretical model (section 2.2) the authors make the following an assumption: "...there is no need to consider the interlayer coupling effect". This simplification must be better explained since the interlayer coupling is one of the main drawbacks of additive manufacturing. Which error is due to this assumption?
- In section 3.2, additional information must be provided about the dynamic experiment (sampling rate, filtering, detailed sensor data, etc.)
- The same for the simulation carried out for obtaining the predicted values of frequency response. Which software, resolution, etc.?
- In section 3.4 errors are around 10-13%. Are these values good for considering that the theoretical model is reliable? Justify, please.
Finally and very important. The paper repeats the same concepts included in the following paper of the same authors:
Shock and Vibration, Volume 2019, https://doi.org/10.1155/2019/121943, Experimental and Theoretical Analysis on the Dynamic Characteristics of Fused Filament Fabrication Plates
This information should not be reproduced again in this paper.
Reviewer 2 Report
The paper might contain some new and significant scientific information adequate to justify publication.
However the reporting of the study need to be improved. The reporting of the experimental methods and results should be more complete and accurate.
Abstract does not contain any quantitate data or clear results, only method.
In AM I recommend to use ISO / ASTM52900 – 15 Standard Terminology for Additive Manufacturing – General Principles – Terminology. For the sake of clarity and for future understandability and indexing when standard name overrules other. Example FFF should be material extrusion.
Literature part there is references related to other process than material extrusion. Such as Ti6Al4V. Those should be removed. Please give reference for simple post processing – since it is not simple if the part is complex or need to be accurate in any AM process. Industry seems Accuracy levels and tolerances one of the biggest barriers for adoption.
Source: "Evaluating the Readiness Level of Additively Manufactured Digital Spare Parts: An Industrial Perspective." Applied Sciences 8.10 (2018): 1837.
Also more sources for dynamic behavior such as:
Source: "Mechanical and Dynamic Behavior of Fused Filament Fabrication 3D Printed Polyethylene Terephthalate Glycol Reinforced with Carbon Fibers." Polymer-Plastics Technology and Engineering 57.16 (2018): 1715-1725.
Overview of mechanical properties:
Source: "Characterization of the mechanical properties of FFF structures and materials: a review on the experimental, computational and theoretical approaches." Materials 12.6 (2019): 895.
Improving the surface quality and faults in material extrusion process by coating:
Source: Kestilä, Antti, et al. "Towards space-grade 3D-printed, ALD-coated small satellite propulsion components for fluidics." Additive Manufacturing 22 (2018): 31-37.
What PLA was used? Why you did not choose example ABS which is more relevant to industry?
How about the post processing of the parts? How was the supports removed or was there any? What kind of finishing was used if used?
About Mechanical characterization. Amount of samples is quite small. How much variation there was between measurements? What was the repeatability?
Explain in detail how the experiments of the samples have been planned and analyzed (factors, levels, type of experimental plan, replications, analysis of variance and related statistical tests. Was there predefined experimental design, e.g. a factorial plan? Explain how the process variables(if tested) influence the responses (individual effects and possible interactions).
About the orientation of parts, what was the printing times and estimated costs for parts? For material extrusion the cost varies remarkable based orientation of the parts.
Source: "Effect of build orientation in 3D printing production for material extrusion, material jetting, binder jetting, sheet object lamination, vat photopolymerisation, and powder bed fusion." International Journal of Collaborative Enterprise 5.3-4 (2016): 218-231.
Comparison for values of “normal” PLA is needed in each phase.
I recommend Major revision for the paper to see more about the details and novelty of the submission.
Round 2
Reviewer 2 Report
Requested modifications seems to be made – the paper can be accepted.